# Oral Manifestations and Complications in Childhood Acute Myeloid Leukemia

**DOI:** 10.3390/cancers12061634

**Published:** 2020-06-19

**Authors:** Francisco Cammarata-Scalisi, Katia Girardi, Luisa Strocchio, Pietro Merli, Annelyse Garret Bernardin, Angela Galeotti, Fabio Magliarditi, Alessandro Inserra, Michele Callea

**Affiliations:** 1Pediatrics Service, Regional Hospital of Antofagasta, Antofagasta 12440, Chile; 2Department of Hematology/Oncology, Cell and Gene Therapy, Bambino Gesù Children’s Hospital, 00165 Rome, Italy; katia.girardi@opbg.net (K.G.); luisa.strocchio@opbg.net (L.S.); pietro.merli@opbg.net (P.M.); 3Unit of Dentistry, Bambino Gesù Children’s Hospital, 00165 Rome, Italy; annelyse.garret@opbg.net (A.G.B.); angela.galeotti@opbg.net (A.G.); fabio.magliarditi@opbg.net (F.M.); 4Pediatric Surgery, Bambino Gesù Children’s Hospital, 00165 Rome, Italy; alessandro.inserra@opbg.net

**Keywords:** leukemia, acute myeloid leukemia, oral manifestations, treatment

## Abstract

Acute myeloid leukemia (AML) is a heterogeneous group of diseases, whose classification is based on lineage-commitment and genetics. Although rare in childhood, it is the most common type of acute leukemia in adults, accounting for 80% of all cases in this age group. The prognosis of this disease remains poor (especially in childhood, as compared to acute lymphoblastic leukemia); however, overall survival has significantly improved over the past 30 years. The health of the oral cavity is a remarkable reflection of the systemic status of an individual. Identification of the signs and symptoms of oral lesions can act as a warning sign of hidden and serious systemic involvement. Moreover, they may be the presenting feature of acute leukemia and provide important diagnostic indicators. Primary oral alterations are identified in up to 90% of cases of acute myeloid leukemia and consist of petechiae, spontaneous bleeding, mucosal ulceration, gingival enlargement with or without necrosis, infections, hemorrhagic bullae on the tongue, and cracked lips. Poor oral hygiene is a well-known risk factor for local and systemic infectious complications. Oro-dental complications due to AML treatment can affect the teeth, oral mucosa, soft and bone tissue, and contribute to opportunistic infections, dental decay, and enamel discoloration. The treatment of acute myeloid leukemia is still associated with high mortality and morbidity. The management is multimodal, involving aggressive multidrug chemotherapy and, in most cases, allogenic bone marrow transplantation. Periodontal and dental treatment for patients with leukemia should always be planned and concerted with hematologists.

## 1. Introduction

Leukemia is a heterogeneous group of hematological disorders arising from hematopoietic stem cells [1], resulting from the uncontrolled proliferation of neoplastic cells [2,3], characterized by impaired differentiation [1,2] and programmed cell death [2]. The failure of maturation of precursor cells results in the accumulation of blasts in the bone marrow with consequent suppression of normal hematopoiesis, leading to deficiency of mature leukocytes, erythrocytes, and platelets [4]. Life-threatening complications are represented by infections, frequently recurrent, as well as severe bleeding episodes [1,2,3,5]. Leukemic cells can invade various organs: Liver, spleen, central nervous system (CNS), bone, and the gingiva. Gingival infiltration can be demonstrated by biopsy [1,3].

Leukemia is the most common form of pediatric cancer in children younger than 15 years old [3,6]. Worldwide incidence is 3.7 per 100,000 and accounts for about 4% of all deaths from malignancies [4]. According to clinical behavior, leukemia is distinguished into acute and chronic [1,5]. Acute leukemia is abrupt in onset and aggressive [4], and the primitive “blast” cells are released into the peripheral circulation, whereas in chronic leukemia, cells tend to be more mature, with normal morphologic characteristics and function when released into the circulation [2]. According to the lineage of the origin of blasts, leukemia may be classified as lymphoblastic or myelogenous. Both may occur in an acute or chronic form and at different ages. Acute lymphoblastic leukemia (ALL) is typical of the pediatric age, whilst acute myeloid leukemia (AML) is more common in adult age. A subgroup of myelogenous leukemia, frequently involving the oral cavity, is monoblastic leukemia [2,5,7].

This review is aimed at emphasizing the oral manifestations and complications of AML resulting from the underlying disease process and following treatment.

## 2. Acute Myeloid Leukemia

Acute myeloid leukemia (AML), also known as myelogenous or myeloblastic leukemia [8], is a highly aggressive malignant disease, representing approximately 25% of pediatric leukemia [9]. The mortality rate is high, highlighting the essential need for an accurate and rapid diagnosis [10,11]. The diagnostic pathways are comparable to adults and include morphology (peripheral blood/bone marrow biopsy), cytochemistry, immunophenotyping, and specific molecular genetics. The results can predict treatment response and risk assessment [1,8,12].

The etiology of AML is poorly defined [1]. Genetic disorders, previous chemotherapy/radiation exposure [1,5], myelodysplastic syndromes [1,5], and exposure to carcinogenic chemicals have been identified as risk factors [5]. Only a small proportion of children and adolescents develop AML as part of hereditary syndrome, and most frequently transient leukemia or myeloid leukemia is seen in children with Down syndrome [1,9]. However, more than 250 gene mutations or other chromosomopathies, including Turner syndrome, damage-associated DNA repair defects, such as Fanconi anemia and Bloom syndrome, can predispose to pediatric myeloid malignancies [13,14]. Recently, acute leukemia, both lymphoblastic and myeloid in adults [15,16], and myeloid in a child [17] have been reported in cleidocranial dysplasia (CCD) associated with *RUNX2* gene mutations. *RUNX1* mutation is the most frequent gene alteration in AML [17]. The occurrence of CCD in patients with a mutation in the *RUNX1* gene appears to be particularly relevant for two reasons: (a) Blood malignancies demonstrating mutations at the equivalent residues RUNX1-2 suggest a common leukemogenic pathway [17], (b) *RUNX2* is an important gene in osteoblastic activity and in teeth development process [18].

AML progresses rapidly and is typically fatal within weeks or months if left untreated [1]. Life-threatening complications result in recurrent infections and also severe bleeding episodes [1,2,3,5]. The treatment-related mortality (TRM) of AML is decreasing due to the introduction of new drugs, improved prognostic factors, and risk group stratification and modern treatment protocols that include intensive induction chemotherapy, followed by post-remission treatment: Additional anthracycline and high dose cytarabine or allogenic hematopoietic stem cell transplantation (HSCT) for subgroups at high risk of recurrence. Supportive care (antibiotic prophylaxis/treatment, new antifungal agents) and Intensive Care Unit (ICU) support have largely reduced morbidity and mortality.

The World Health Organization classification divides AML into six subtypes:(1)AML with recurrent cytogenetic translocations;(2)AML with myelodysplasia-related changes;(3)Therapy-related myeloid neoplasms;(4)AML not otherwise specified (NOS);(5)Myeloid sarcoma;(6)Myeloid proliferations related to Down syndrome [19].

AML can be further classified, according to morphological criteria, using the French–American–British (FAB) classification, that commonly classifies into 8 subgroups as: M0 undifferentiated leukemia, M1 acute myeloblastic leukemia, M2 acute myeloblastic leukemia with maturation, M3 acute promyelocytic leukemia, M4 acute myelo-monocytic leukemia, M5 acute monocytic leukemia, M6 acute erythroblastic leukemia, and M7 acute megakaryoblastic leukemia [19].

Pediatric AML is a relatively rare malignancy that has benefited from multicenter clinical trials from international cooperative groups (such as Associazione Italiana Ematologia e Oncologia Pediatrica (AIEOP)–Berlin Frankfurt Munster (BFM), Nordic Society for Pediatric Hematology and Oncology (NOPHO), etc.) [12]. Therefore, the complete remission (CR) rate is around 80–90%, while the relapse incidence is 30–40%; overall survival (OS) and event-free survival (EFS) approach 70% and 50%, respectively [20]. A comprehensive review of current treatment strategies (including risk-stratification, chemotherapy courses, HSCT use, and new drugs including biologics) for de novo and relapsed AML has been recently reported [12].

## 3. Oral Manifestations

The health of the oral cavity health significantly reflects the health of the whole organism. Identification of the signs and symptoms of oral lesions can act as a warning sign of hidden and serious systemic involvement [21]. Oral lesions may be the presenting feature of acute leukemia and therefore can be an important diagnostic indicator [5]. Not infrequently, undiagnosed cases of leukemia refer to the dentist with complaints related to oral lesions [1], whose recognition can lead to the diagnosis of AML [21].

Indeed, oral manifestations occur in most patients with AML and are often the first presentation. Although not specific, they can direct a diagnosis of an underlying leukemia, especially in the presence of different lesions, timing, and size at onset [1,2,3]. Still, there is a limited number of studies reporting the prevalence of periodontal status and parameters in AML patients [1,3,10]. However, the oral manifestations are far more common in myeloid and monocytic/monoblastic leukemia [2,5]. Oral changes can also occur in chronic leukemia [1,2]; they differ from those seen in acute leukemia [4] and are not considered specific.

Oral manifestations of leukemia include petechiae or spontaneous bleeding in 56% of patients [1,2,5,22], mucosal ulceration in 53% [1,2,3,4,5], and gingival enlargement in 36% [1,6,11,22,23], with or without necrosis [5]. These features are the most common initial diagnostic manifestations of leukemia. In addition to mucosal pallor [4], higher caries prevalence [11], herpetic opportunistic infections and candidiasis [3,4,5], temporomandibular joint arthritis, and osteolytic lesions in the mandible may arise [3]. Other oral signs such as palatal pigmentation [4], tooth pain and mobility [24], hemorrhagic bullae on the tongue, cracked lips, parotid swelling [4], and chin numbness were less commonly reported [2]. All these oro-dental features, summarized in Table 1 and Figure 1 and Figure 2, show common oro-dental manifestations in AML patients.

A large series reported that the most common oral signs of leukemia occurring after diagnosis are oral bleeding or purpura. Notably, some authors have found that patients with acute leukemia displaying these findings in the initial or post diagnostic periods tended to have shortened survival time as compared to patients who did not show these lesions [25].

### 3.1. Gingival Alterations

Gingival infiltration represents the initial manifestation of AML in 5% of cases, and is more frequently seen in myelomonocytic (M4) and monocytic (M5) leukemia. The proposed hypothesis for gingival involvement is based upon the consideration of its microanatomy and of the expression of endothelial adhesion molecules which allows infiltration of leukocytes [2], leading to an overgrowth with a soft consistency [8]. Dreizen et al. [26], in an observational study on 1076 adults hospitalized for chemotherapy, reported gingival involvement in 66.7% of AML-M5 and 18.5% of M4 patients, respectively. Gingival lesions were particularly prone to infectious complications in patients with poor oral hygiene. 

Gingival enlargement and ulcerations may be due to either neutropenia or direct infiltration of immature (blasts) proliferating leukocytes, or be secondary to thrombocytopenia and immunodeficiency [1,5,23]. This infiltration leads to an increase in gingival thickness and formation of pseudo-pockets, resulting in secondary inflammatory infiltration [1]. The continuous trafficking of myeloid cells in specialized post-capillary venules accounts for egress of these cells from the circulation into the tissues at the sites of gingivitis or periodontitis [8]. Leukemic gingival infiltration is not observed in edentulous individuals, suggesting that local irritation and trauma of the teeth are associated in the pathogenesis [2,26]. The gingival findings are reported to be partially dependent on the inflammation of the tissues [4].

Despite being rarely reported in the literature [1], gingival enlargement may be the first manifestation of acute leukemia, especially in AML [1,8,11,23]. The overgrowth of gingiva is characterized by an accumulation of connective tissue with the presence of an increased number of cells. According to the etiologic factors and pathologic changes, gingival enlargement can be classified as inflammatory, drug-induced, neoplastic, false enlargement, and is often associated with systemic diseases (in particular, granulomatous diseases and leukemia) [1,23].

Differential diagnosis of leukemic gingival enlargement with gingival enlargement due to different etiologies is important because of the lethal and poor clinical outcome of the former [21]. Regression of gingival enlargement has been observed within three to four weeks from the initiation of chemotherapy regimens [1]. It has to be noted that, although rarely, gingiva can be a site of extra-medullary localization (also at relapse) [27] (named myeloid sarcoma, occurring in 3–5% of patients with AML and involving more frequently the skin, the bones, or the gastrointestinal tract).

### 3.2. Oral Hygiene and Risk of Infections

Poor oral health (as defined by the presence of gingivitis/periodontitis) is a predictor of increased risk of infectious complications in hospitalized leukemic patients during chemotherapy [28,29]. Poor hygiene is a well-known risk factor for leukemic gingival overgrowth, destructive periodontal disease [8], and for tissue necrosis, predisposing to oral pain, bleeding, and super-infections. Advanced cases may also present with malaise, fever, laryngeal pain, and cervical lymphadenopathy [4].

In patients showing high levels of oral hygiene, the gingival overgrowth tends to be mild, especially with respect to mechanical tooth cleaning. The oral cavity is a major cause of sepsis, because inflamed gingival tissues serve as a major entryway for bacteria and bacterial products, such as endotoxins, which elevate the serum levels of inflammatory cytokines and pyrogenic mediators, for example, interleukins 1 and 6. In addition to sepsis, local infections may give rise to abscesses. A study carried out in 73 young adults with a diagnosis of AML reported that around three-quarters had either fair or poor oral hygiene. A statistically significant association between dental plaque levels and both gingival overgrowth and periodontal index (*p* < 0.001) was observed [8]. More than one-third of patients have been reported with significant or life-threatening infections, most of which were of bacterial origin [8]. Djuric and colleagues [30] conducted a randomized controlled trial on 34 patients hospitalized for induction therapy for acute leukemia (most of them had AML). They were randomly assigned to a) intensive dental care protocol (including dental treatment, plaque and calculus removal prior to chemotherapy, and supervised oral hygiene measures during chemotherapy) or b) no specific care. The authors found that 44% showed significant improvement in the oral hygiene and gingival indices. Moreover, a reduction in the number of non-favorable microorganisms (such as *Candida albicans* and gram-negative bacilli) in the intensive dental care group was observed. During the whole examination period, intensive dental care patients group developed less severe and less painful oral complications compared to the limited dental care patient group. The authors therefore recommended that proper dental care and preventive measures both before and during chemotherapy can be beneficial and should be offered to these patients [30].

Moreover, reduced salivary flow induces changes in the bacteria colonizing the oral cavity and promotes caries-related microflora. Another consequence of hyposalivation and dry mouth is the excessive use of products with sugar, a further risk factor for caries development [31].

## 4. Treatment

The treatment of AML is challenging and characterized by high mortality and morbidity [12]. The management is multimodal, involving aggressive multidrug chemotherapy and allogeneic bone marrow transplantation in many patients [2,32]. Adverse effects of oncological treatments are unavoidable in AML patients and can impact the oral health status, thus significantly affecting the quality of life of the survivors [3,31,32]. Malignant cells are the target of antineoplastic drugs, but the oral epithelium and other cells with high mitotic rates are usually affected by the treatment. The adverse effects of chemotherapy and conditioning regimens for HSCT (in few selected patients) [3] depend on the type and dosage, as well as the age of the patient at the beginning of treatment.

Gingival hyperplasia can resolve completely, or at least partially, with effective chemotherapy within 3 to 4 weeks [1,33,34]. Therefore, these patients may require a scheduled careful preventive program, long-term follow up, with pre-emptive treatment with the objective to minimize the consequences of the disease, and chemotherapy [32]. 

Periodontal and dental evaluation and treatment for patients with leukemia should always be planned and concerted with the hematologists. Daily plaque removal from the teeth can resolve gingival inflammation [35]. If systemic conditions allow periodontal debridement (scaling and root planing), the patients should be treated with antibiotics prophylaxis. Twice daily rinsing with 0.1–0.2% chlorhexidine gluconate is recommended after oral hygiene [2,36]. This can minimize oral complications during remission-induction chemotherapy, demonstrating a superior oral health and prophylaxis of oral candidiasis in the myelosuppressed patient [35]. Periodontal surgery should be avoided until remission [2]. Abscesses require antibiotic treatment to relieve pain; drainage can be performed under antibiotic treatment and paying attention to hemostasis. Caries may be easily treated in the dental office if superficial (involving solely the enamel); however, if the caries extends to the dentin and pulpa, the use of conscious sedation may be used in order to obtain a better collaboration of the patient. Sometimes deep sedation and general anesthesia may be required in non-cooperative patients or in extensive caries involving most deciduous and/or permanent teeth, hence requiring a one-step treatment, often a few weeks before the HSCT or Bone marrow Transplantation (BMT).

The use of antifungal prophylaxis, such as posaconazole formulations, is necessary in high-risk patients [37]. The prevention of invasive fungal infections is important in patients with AML receiving cytoreductive chemotherapy. In a multicenter study involving 5517 patients, the use of voriconazole was significantly associated with a reduction in a proportion of patients switching to intravenous antifungal agents as compared to the use of first-generation azole (−21.0% (95% confidence interval [CI] −33.4 to −8.6)). This effect was stronger in patients aged < 65 years than in those aged ≥ 65 years (−40.6%, 95% CI −63.2 to −17.9; −21.9%, 95% CI −35.8 to −8.1, respectively) [7].

The management of oral health of AML patients is guided by pre-existing gingival and periodontal pathology like gingival infiltration, spontaneous gingival bleeding, or gingival erythema [8]. However, certain complications of chemotherapy and radiation such as hemorrhage, xerostomia, mucositis, and recurrent herpes simplex virus type 1 infection should be identified, and the treatment plan modified [4]. For example, swallowing may become difficult due to the treatment related mucositis, or ulcers caused by herpes simplex virus type 1 infection and oral bleeding [31]. The evaluation of the periodontal status is necessary before treatment to avoid local and systemic complications. Together with a significant reduction in gingival inflammation and maintenance of the periodontal health, a remarkable decrease in the incidence and severity of oral mucositis was observed [38]. Palifermin is a medication frequently used for mucositis prevention in patients after allogeneic hematopoietic stem cell transplantation [39]. The use of honey mouthwash can reduce the incidence and severity of mucositis and reduce or eliminate the possibility of weight loss in these patients [40].

In the post-transplant period, graft versus host disease (GVHD) represents a most severe complication that can cause, amongst others, oropharyngeal fibrosis that may lead to dysphagia. Photobiomodulation therapy can alleviate the symptoms by reducing tissue fibrosis via a putative anti-fibrotic role of TGF-β [41].

Dental alterations are identified in up to 90% of cases of AML (Table 2). Associations with the use of anti-neoplastic regimens are described in Table 1 [3,31]. Equally, these anomalies depend on the odontogenetic phase during which the therapy was conducted [3]. In pediatric leukemic survivors, the decayed, missing, or filled permanent teeth index is important. The negative impact of cytostatic drugs on the oral mucosa, as well as the poor oral hygiene during the treatment phase, are the main causes of oral health decline [31]. Therefore, the high prevalence of oral diseases supports the need for an early and consequent oral treatment in leukemia patients, especially considering the subsequent therapy [42]. The cost of AML treatment is substantially increasing due to the diverse options and new expensive oral therapies and varies depending on the approach and the country studied [43].

### The Role of Dentists in Diagnosis and Treatment

Dentists play an important role in detecting oral signs which are suspicious or diagnostic of leukemia in early stages during inpatient or outpatient consultation, and subsequently in referring cases to hematologists for hematological examination. Patients should also be referred to the Oral Health Program clinic where imaging, precision periodontal instruments, or any other auxiliary methods to evaluate oral tissue are available [44]. Dentists are part of the multidisciplinary team during the pre-treatment and post-treatment period of AML.

In hospitalized patients, stomatologists usually perform intraoral examination at the bedside with the aid of a frontal light emitting diode light while respecting biosafety guidelines and principles of oral semiology. Dental hygiene, calculus stone removal, filling of caries, exfoliating or decayed teeth extraction, and discoloration enamel can be treated, in the case of non-transportable children, at the bedside with the aid of portable dental units (Figure 1 and Figure 2). The steps to evaluate the oral mucosa are standardized in the following order: Vermillion lip border, labial mucosa, buccal mucosa, tongue (dorsum, lateral, and ventral surface), hard and soft palate, and oropharynx on both sites of the mouth. Oral conditions detected during examinations are grouped into those related to the underlying disease or its treatment (dry lips or mucosa, mucositis, bleeding disorders, fungal or viral infections, and aggravation of odontogenic infections), and those unrelated (friction oral keratosis, traumatic ulcer, reacting fibrous hyperplasia, leukoplakia, aphthous ulcer, coated tongue, mucocele). Preexisting constitutional or syndromic dental abnormalities are recorded and their treatment suspended or postponed after the remission/recovery.

## 5. Long-Term Adverse Effects

It is well known that that anticancer therapy in the pediatric age may affect tooth development. Several studies showed that the administration of anthracyclines, a class of chemotherapeutic agents commonly used for the treatment of AML, resulted into disturbed odontogenesis [45,46]. However, the precise molecular mechanisms through which chemotherapy or radiotherapy induce dental aberrations have not been elucidated. It has been hypothesized that this effect can be due either to a direct toxic effect toward the odontogenic cells, or to the interference with the signaling network between ectoderm and mesenchyme or within one tissue layer [45,46]. 

As a matter of fact, cancer survivors present dental abnormalities more frequently observed than in healthy patients. Indeed, Proc and colleagues reported that the more frequent abnormalities observed were agenesis, hypodontia, microdontia, and teeth with short roots. In this cohort of patients, dental anomalies in cancer survivors occurred more frequently in some tooth groups (e.g., microdontia was most often found among second premolars and second molars) and were not observed in others [47]. Notably, in this cohort there was no correlation between the frequency of these anomalies and the age at the beginning or termination of chemotherapy. It has been hypothesized that the unexpected lack of relationship between the distribution of missing teeth and the age of the child at the beginning of the therapy could be due to: i) The presence of a genetic background favoring tooth agenesis, or ii) the fact that the tooth buds were completely destroyed by chemotherapy regardless of their stage of development. 

Conversely, other studies found a correlation between the age at the beginning of chemotherapy/radiotherapy treatment and dental abnormalities [48]. Indeed, Hölttä and coauthors observed that, in a cohort of 55 pediatric patients receiving an allogeneic hematopoietic stem cell transplantation (HSCT), the recipient age at the time of transplantation was found to have a negative correlation with the number of missing teeth and microdontic teeth, with an age at HSCT < 3 years of age being associated with the highest incidence of abnormalities. Moreover, the use of total body irradiation (TBI) as part of the conditioning regimen before stem cell infusion appeared to have little or no effect on the prevalence of tooth agenesis or on the prevalence of microdontia. TBI, while commonly used for the preparative regimen of children affected by ALL, is less used for AML pediatric patients; however, recent findings in T cell-depleted HSCT from an haploidentical donor suggest that it can be useful to lower the incidence of relapse also in this kind of disease [49]. Similarly, the Danish Registry of Childhood Cancer found that, in a cohort of pediatric cancer survivors (most of them affected by acute leukemias), the earlier the exposure, the more frequent was microdontia of premolars or permanent molars [50]. The age dependence of abnormalities secondary to the treatment reflects the embryology and the development of teeth that is a stepwise process in which a series of genes are involved in a sequential or interacting modality [18].

Physicians caring for long term pediatric cancer survivors should always keep in mind that odontogenesis disturbances have a relevant psychological impact, impairing the quality of life of teenagers and young adults; especially for enamel defects. Indeed, a study conducted on 547 pupils (aged 11–15 years) who were given full-face photographs of a boy and girl without an enamel defect, showed that young people may make negative psychosocial judgments on the basis of enamel appearance [51].

## 6. Conclusions

Oral lesions, especially gingival hyperplasia, may be the first signs of hematologic diseases. Therefore, oral health-care professionals should be aware of the oral manifestations in AML, and of the importance of detecting the signs related of this systemic condition that often compel the patient to seek for dental care first [44]. To establish an early diagnosis and subsequent management, the dental practitioner must clearly recognize these alterations and investigate them with additional tests and further refer the patient to a specialized center [4]. Patients with AML can undergo a series of oral complications, most of them secondary to the aggressive chemotherapy. Oral hygiene is a most important measure to prevent infections that are at risk for abscess formation or sepsis. Moreover, oral health can be maintained with cooperation between pediatricians, pediatric oncohematologists, hematologists, dental surgeons, and dental hygienists [28]. The role of the dentist in a multidisciplinary team is essential also for prevention, diagnosis, and control of preexisting problems such as tooth decay, periodontal diseases and other alterations of the oral and perioral tissues [44]. In particular, as recommended by the Children’s Oncology Group Guidelines (http://survivorshipguidelines.org/pdf), dental examination should be performed twice yearly as part of the medical follow-up by monitoring late effects in long-term survivors of childhood cancer carried out by a dental surgeon experienced in survivorship care.

## Figures and Tables

**Figure 1 cancers-12-01634-f001:**
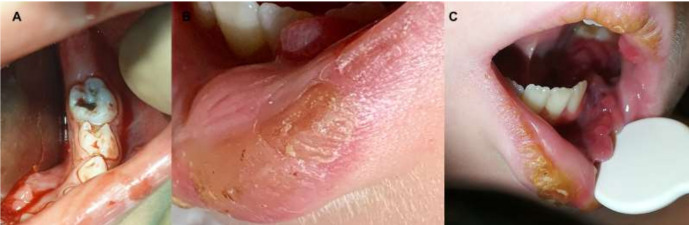
(**A**) Caries in the lower primary molars, intraoral bleeding, and gingival enlargement/hyperplasia. (**B**) Cracked lips and gingival enlargement. (**C**) Cracked lips, gingival enlargement/hyperplasia, and buccal bleeding.

**Figure 2 cancers-12-01634-f002:**
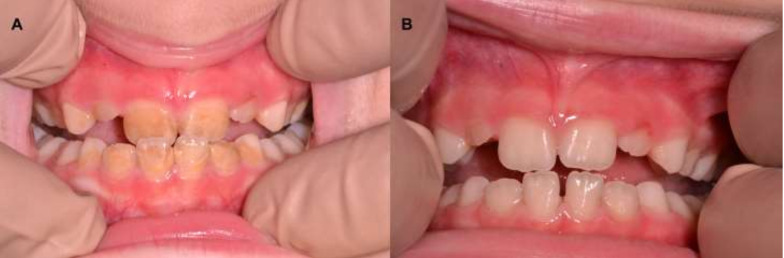
(**A**) Enamel discoloration and presence of calculus stone. (**B**) Picture showing intraoral view of patient shown in Figure 2A after removal of the discoloration and calculus stone in a few sessions of scaling and polishing.

**Table 1 cancers-12-01634-t001:** Soft and hard tissue alterations in acute myeloid leukemia (AML).

Findings
Petechiae
Spontaneous bleeding
Mucosal ulceration
Gingival enlargement with or without necrosis
Mucosal pallor
Enamel discoloration
Herpetic opportunistic infections
Candidiasis
Temporomandibular joint arthritis
Osteolytic lesions in the mandible
Palatal pigmentation
Tooth pain and mobility
Hemorrhagic bullae on the tongue
Cracked lips
Parotid swelling
Chin numbness
Caries

**Table 2 cancers-12-01634-t002:** Dental alterations associated with the use of anti-neoplastic therapies.

Enamel Malformation	Discoloration
Radicular anomalies	Hypoplasia
	Resorbed or tapered roots
	Early apical closure
	Delayed dental development
	Dental impaction
Dental shape anomalies	Microdontia
	Macrodontia
	Taurodontia
Anomalies in numbers	Hypodontia
	Supernumerary teeth

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
