# Peer review of "Oral Manifestations and Complications in Childhood Acute Myeloid Leukemia"

_cancers, 2020, doi:10.3390/cancers12061634_

Round 1

Reviewer 1 Report

The authors have addressed by previous suggestions.

Reviewer 2 Report

The authors have considered all relevant aspects in the new version. 

This manuscript is a resubmission of an earlier submission. The following is a list of the peer review reports and author responses from that submission.

Round 1

Reviewer 1 Report

  • The title should clarify that the article focuses as well on the manifestations of AML in the oral cavity as on the consequences of AML therapy for the orodental field. 
  • The authors may add a couple of illustrative photographic pictures demonstrating typical complications in the oral cavity (e.g. gingival hyperplasia, buccal hemorrhage, progressive caries…). 
  • The general aspects on AML may be shortened. The authors should focus the manuscript very clearly on the aspects referring to their topic. 
  • Altogether, the manuscript closes a gap in the field of AML for clinical hematologists and oncologists.  

Reviewer 2 Report

The authors have written a review on oral manifestations of AML but have also touched on issues related to dental complications and their treatment. Overall, this is a very niche topic and there is minimal data available in the literature on which to base such a review. Therefore, much of this is expert opinion, which unfortunately lead to a lot of generic, non-data-driven statements. I have the following specific comments:

  1. Title: Should be more specific about the scope of the review. For example,  "manifestations and complications" I am not sure how much the authors have provided abut therapeutic options in the review. 
  2. Abstract: Transplant is offered in "many" but not "most" cases.
  3. General: This seems to be coming from the perspective of pediatric oncology. This could be stated more explicitly, particularly when the authors are describing treatment or prognosis of AML, as these vary significantly between populations.
  4. General: The reference numbers are not synced with the reference list. This must be corrected.
  5. Sections 1 (Intro) and 2 (AML) are highly redundant. Most of Introduction could be removed.
  6. Section 2: I do not understand the last sentence of first paragraph. 
  7. Section 2: I am not aware of any good data that EBV or other viruses predispose to AML. This should be removed.
  8. Section 2, paragrph 4: Should be "AML" not "ALM"
  9. Section 2: The WHO categories are outdated. There are six categories as of 2016 (Arber et al. Blood 2016)
  10. Section 2, last paragraph: When discussing incidence of AML and ALL it should be clear the authors are discussing pediatric.
  11. Section 3: Please provide citation that oral manifestations occur in 2/3 of all cases.
  12. Section 3: The mention of case report of HPV-related SCC is so anecdotal that it should be removed. Is there any evidence to suggest this was a true association and not just coincidence.
  13. Section 3: The figure does not add anything to manuscript and should be removed.
  14. Section 3: The citation that dental lesions are associated with shortened survival was published before the era of modern therapy. This claim is therefore dubious and I suggest to remove.
  15. Section 3.1: I would highlight that gingival enlargement is more common with myelomonocytic leukemia.
  16. Section 3.1: Partial or complete regression of gingival enlargement can be seen much sooner than after 4 weeks. The authors mention different numbers elsewhere in the manuscript.
  17. Section 3.2: I do not understand the statement that 44% showed improvement in oral hygeine. 44% of what patients? (there were 2 groups)
  18. Section 3.2: The authors should mention dental abscess as complication. Should also be mentioned in treatment. How should these patients be treated and monitored. When can dental procedures be performed?
  19. Section 6: I do not know by what data the authors state that dentists diagnosis in AML in 25-33% of cases.

Reviewer 3 Report

The topic of oral manifestations of AML would be interest to many who treat AML on daily basis. However, I suggest authors to restructure this paper extensively. Also, paper discusses not only AML, but dental complications from most intensive chemotherapies, adult versus pediatric, or both, then conclusion ends with recommendation from pediatric oncology group, how about adults?

My specific comments are below

Line 46: “leukemia is classified into chronic, subacute and acute….”, leukemias are either acute or chronic, there is no subacute leukemia

Line 54: correct punctuation error

Line 55-116: I suggest authors eliminate this section, because they try to summarize (partially) the whole topic of AML (epidemiology, etiology, clinical presentation, inherited syndromes, disease classification, treatment), bits and pieces from all. No need for that, review should focus soft tissue involvement in AML.

Line 118: please rephrase this sentence, not clear [systemic status?]

Line 123: repetition

Line: 132- 134: it would be really nice if you could put real life images of these manifestations.

Figure 1: not sure how this figure would contribute to this review, rare condition

Lines 132-140: suggest to make a table of these symptoms

Line 151: reference

Line 333:” activating the diagnosis..”, not sure what that means